# Sorption and Desorption Studies of Pb(II) and Ni(II) from Aqueous Solutions by a New Composite Based on Alginate and Magadiite Materials

**DOI:** 10.3390/polym11020340

**Published:** 2019-02-15

**Authors:** Keltoum Attar, Hary Demey, Djamila Bouazza, Ana Maria Sastre

**Affiliations:** 1University of Oran 1 Ahmed Ben Bella, Laboratory of Chemistry of Materials, B.P 1524 El M’naouer Oran, Algeria; attar_keltoum@hotmail.com (K.A.); bouazzadjamila99@gmail.com (D.B.); 2Department of Chemical Engineering, Universitat Politècnica de Catalunya, EPSEVG, Av. Víctor Balaguer, s/n, 08800 Vilanova i la Geltrú, Spain; 3Commissariat à l’Energie Atomique et aux Energies Alternatives, CEA/DRT/LITEN/DTBH/STBH/L2CS, 17 rue des Martyrs, 38054 Grenoble, France; 4Department of Chemical Engineering, Universitat Politècnica de Catalunya, ETSEIB, Diagonal 647, 08028 Barcelona, Spain

**Keywords:** alginate, heavy metal, H-magadiite, organophosphorus extractants, sorption, desorption

## Abstract

A new composite material based on alginate and magadiite/Di-(2-ethylhexyl) phosphoric acid (CAM-D2EHPA) was successfully prepared by previous impregnation of layered magadiite with D2EHPA extractant, and then immobilized into the alginate matrix. Air dried beads of CAM-D2EHPA were characterized by FTIR and SEM–EDX techniques. The sorbent was used for the separation of lead and nickel from nitrate solutions; the main parameters of sorption such as contact time, pH of the solution, and initial metal concentration were studied. The beads recovered 94% of Pb(II) and 65% of Ni(II) at pH 4 from dilute solutions containing 10 mg L^−1^ of metal (sorbent dosage, S.D. 1 g L^−1^). The equilibrium data gave a better fit using the Langmuir model, and kinetic profiles were fitted using a pseudo-second order rate equation. The maximum sorption capacities obtained (at pH 4) were 197 mg g^−1^ and 44 mg g^−1^ for lead and nickel, respectively. The regeneration of the sorbent was efficiently carried out with a dilute solution of HNO_3_ (0.5 M). The composite material was reused in 10 sorption–elution cycles with no significant differences on sorption uptake. A study with synthetic effluents containing an equimolar concentration of both metals indicated a better selectivity towards lead ions.

## 1. Introduction

The removal of pollutants (such as metal ions) from water is currently an important environmental concern. Heavy metals are particularly toxic and carcinogenic; their presence in the aquatic ecosystems poses human health risks due to their non-degradable and persistent nature. The presence of these contaminants in the environment is mainly due to industrial activities, such as metal plating, oil refining, petrochemical, dyeing, painting, mining, and fertilizers/pesticides production [1].

Lead and nickel are among the most toxic metals released into the environment, according to the World Health Organization. The recommended limit concentrations were set in the guidelines for drinking water as 0.01 mg L^−1^ and 0.07 mg L^−1^ for Pb(II) and Ni(II), respectively [2]. The common methods used for the removal of these metals are chemical precipitation [3,4], electrochemical treatment [5,6], solvent extraction [7], membrane technology [8,9], ion-exchange [10,11], and sorption [12,13,14,15]. According to Demey et al. [16], all these techniques have intrinsic operation shortcomings; e.g., membrane technology and electrochemical treatment require high energy consumption, which is not cost-effective for high volumes of contaminated water; chemical precipitation involves the disposal and treatment of toxic sludge, whereas solvent extraction produces a secondary effluent due to the use of large amounts of organic solvents. Sorption is considered as the most suitable method for removing heavy metals because of its promising advantages [12,13,14,15]: (i) high efficiency uptake for binding metal ions at low concentrations; (ii) energy saving, and iii) easy recycling of the loaded sorbent. Nevertheless, a good design of the material is required to achieve high performances [17,18,19].

Bouazza et al. [20] agree that the sorption process is non-expensive in comparison with liquid–liquid extraction (in which large volumes of extractant are used). The impregnation of resins was proposed as a technological alternative to a direct solvent extraction technique. Layered silicate materials have been used as supports for impregnation with efficient solvents [7]; the layered crystalline polysilicates are mainly composed of tetrahedral SiO_4_, they usually exhibit a high metal uptake and their manufacturing does not require difficult steps. In recent years, several researchers have studied the adsorption of organic pollutants on modified and unmodified layered silicate [21,22]. Due to their simple manufacture, these materials have become attractive for industrial applications.

Several support-materials have been used for impregnating with extracting agents for metal ion binding; e.g. Berdous et al. [23] used impregnated Amberlite XAD-4 with D2EHPA for the removal of heavy metals from water, the resins based on styrene-divinyl-benzene are effective supports but cannot be considered readily bio-degradable. Vellaichamy [24] investigated the use of multiwall carbon nanotubes impregnated with D2EHPA to remove copper, nickel, lead, zinc, and cadmium from aqueous solutions where the best sorption capacities were obtained at pH 5, and followed the order: Ni(II) > Cu(II) > Pb(II) > Zn(II) > Cd(II). The research developed by Kozlowska [25] studied the transport of heavy metals across cellulose triacetate plasticized membranes containing organo-phosphorous acids; by using D2EHPA, the selectivity order achieved was: Zn(II) > Pb(II) > Cd(II). It is evident that the metal species and the functional groups present in the matrices play an important role in the sorption affinity.

In our previous work [26], a magadiite-based material impregnated with Cyanex 272 was tested as a potential sorbent for cadmium recovery from water. The impregnation of the magadiite support with the extractant was found to increase the selectivity toward cadmium in the presence of nickel ions (at equimolar concentrations). Nevertheless due to the small particle size of the material, it was not possible to evaluate in continuous systems.

It is noteworthy that most sorption studies present in the literature are focused on the use of sorbents in powder form (micro/nano particle size). The small particle size may improve the diffusion properties of the materials and reduce the required contact time on sorption lab-scale assessments, but it could involve several technical problems in industrial applications, such as: low mechanical strength, difficulty in separation from the aqueous phase and high head-loss in column arrangements (continuous system). Demey et al. [16] stated that these drawbacks can be avoided by immobilization of the active powder in biopolymer beads (so-called hybrid materials).

Hybrid ‘low-cost’ sorbents have become a hot topic that has attracted major attention of environmental researchers in recent years. The combination of different matrices with several functional groups may improve the sorption performance and enhance the physical properties of the resulting material [27]. Calcium alginate is a polysaccharide, which is largely used in the preparation of composite sorbents due to the associated versatility, low-cost, and high content in active groups (such as carboxylic and hydroxyl groups) [28]. Some studies found a high affinity of the alginate-based materials for the chelation of heavy metal ions (Cu^2+,^ Ni2^+^, Co^2+^) [29], metalloids (such as boron) [28,30,31] and rare earth elements [32]. Wang et al. [33] studied the sorption of Nd(III) on alginate–silica sorbent (ALG/SiO_2_). It was found that the doped silica can improve the mechanical properties of the beads. Moreover, a raw material based on solid waste containing iron(III) hydroxide was entrapped in a calcium alginate matrix for the sorption of As(V) [34]. The results suggested that the hybrid beads are promising for As(V) removal from waters.

Ngomsik et al. [35] used magnetic alginate beads containing Cyanex 272 for the removal of cobalt and nickel ions from effluents; the sorption results demonstrated that the introduction of Cyanex 272 into the beads increases the selectivity towards cobalt metal. Similarly, Lai et al. [36] tested the performance of four different sorption systems, orange peel cellulose (OPC), banana peel cellulose (BPC), and orange peel cellulose, immobilized in Ca-alginate beads (OPCCA), plus, the banana peel cellulose immobilized in Ca-alginate beads (BPCCA) to recover copper, lead, and zinc ions from aqueous solutions. The authors concluded that the immobilization of the biomass into alginate beads improves the metal sorption uptake.

In this work, H-magadiite impregnated with D2EHPA was immobilized in an alginate matrix for manufacturing beads by ionotropic gelation with a calcium solution. The resulting material (calcium alginate/H-magadiite-D2EHPA: CAM-D2EHPA) was characterized with FTIR, SEM–EDX, and evaluated for the removal of Pb(II) and Ni(II) from aqueous solutions (in both batch and continuous systems). This new composite material, has several advantages compared to the raw H-magadiite powder since it is easy to handle, it can be used in continuous systems to avoid clogging of the columns, it can be reused several times, and the mass loss of the sorbent during several sorption cycles can be significantly reduced due to the rigid encapsulation of the impregnated H-magadiite powder into beads. The regeneration of CAM-D2EHPA beads was evaluated with dilute solutions of HNO_3_ (0.5 M).

## 2. Materials and Methods

### 2.1. Materials

Heavy metal stock solutions were prepared by dissolving lead and nickel nitrate (Pb(NO_3_)_2_ and Ni(NO_3_)_2_·6H_2_O respectively) in Milli-Q water (at constant ionic strength: 0.005 M NaNO_3_) at the required concentrations, the metal salts were provided by Panreac (Barcelona, Spain); solution pH was adjusted with HNO_3_ or NaOH solutions (as required). Silica gel (Ludox 40% *w*/*w*) was used for the H-magadiite preparation, and it was supplied by Sigma-Aldrich (Steinheim, Germany); sodium hydroxide was supplied by Probus (Barcelona, Spain). D2EHPA extractant (98.2% purity) was provided by Sigma-Aldrich (Darmstadt, Germany) and it was used without any further purification. Organic solvents of chloroform (99% purity) and ethanol (96% purity) were obtained from Panreac (Bordeaux-Pessac, France).

The sodium alginate used for the preparation of the composite (CAM-D2EHPA), and the calcium nitrate were supplied by Panreac (Barcelona, Spain).

### 2.2. Preparation of Sorbents

#### 2.2.1. Preparation of H-Magadiite/D2EHPA Material (HM-D2EHPA)

The preparation of H-magadiite was obtained by slow titration of Na-magadiite in suspension with HNO_3_ (1 M) at room temperature for 24 h under mechanical agitation (300 rpm). The detailed process was reported in our previous work [26]. Titration with HNO_3_ up to a pH of 4, is to ensure the exchange of Na ions (from raw Na-magadiite) with the protons of the acid solution [37]. The obtained solid (H-magadiite: HM) was filtered, then washed with distilled water, and dried in an oven at 40 °C for 24 h. Then, the HM material was impregnated with D2EHPA using a dry impregnation mode: (i) D2EHPA was dissolved in 10 mL of ethyl alcohol solution (high purity) and then mixed with 1 g of H-magadiite; (ii) the mixture was stirred overnight (under atmospheric pressure) until complete evaporation of the solvent; (iii) the resulting solid was washed with 0.1 M nitric acid solution to prevent the release of the extracted molecules, and then dried in an oven at 80 °C for 24 h [26].

#### 2.2.2. Preparation of the Calcium Alginate/H-Magadiite-D2EHPA Material (CAM-D2EHPA)

The alginate gel (2% *w*/*w*) was prepared by adding 2 g of sodium alginate in 100 ml of Milli-Q deionized water under stirring (300 rpm) over 3 h, until a homogeneous gel was obtained (stirrer: AGIMATIC-S, Salamanca, Spain). Then, the agitation speed was decreased to 50 rpm for 2 h, to allow the removal of air bubbles from the viscous solution.

CAM-D2EHPA beads were prepared by immobilizing 1 g of HM-D2EHPA in 100 ml of sodium alginate gel (2% *w*/*w*); the suspension was kept under agitation (300 rpm) until the homogenization of the mixture was achieved. Then, the gel obtained was pumped (through a peristaltic pump: MINIPULS3-GILSON, Coueron, France) to form drops with a flow rate of 0.025 L h^−1^, the ionotropic gelation was made in a solution of calcium nitrate Ca(NO_3_)_2_ (1 M). The manufactured beads had an average diameter of 1.8 mm; these were systematically washed with distilled water and air dried for utilization in sorption applications.

### 2.3. Characterization of Sorbents

The identification of the chemical functional groups of the sorbent was determined by Fourier transform infrared (FTIR) analyses; samples of 2 mg of CAM-D2EHPA were studied using a BRUKER IFS 66 FTIR spectrophotometer (Ettlingen, Germany) equipped with a reflection diamond accessory (Platinium ATR); the spectrum interval was registered from 4000–400 cm^−1^.

Morphology and micrographs of CAM-D2EHPA were obtained by SEM–EDX studies using a JEOL JSM 7100 F Schottky Field Emission Scanning Electron Microscope (JEOL Ltd, Peabody, MA, USA). The microscope was equipped with an Energy Dispersive X-ray (EDX) spectrometer (INCA 250, Oxford instruments, Oxford, UK) for chemical analyses. The samples were coated with carbon before observations. The apparatus was operated at 20 keV. SEM–EDX analyses were used to detect the main elements present at the whole solid surface.

The amount of D2EHPA inserted into the H-magadiite was determined using a UV–vis spectrophotometer (model SPECORD-210, Bayern-Munich, Germany) at a wavelength of 254 nm. The metal concentrations were analyzed by using an Agilent 4100 MP–AES Spectrometer (Microwave Plasma–Atomic Emission Spectrometry; Agilent Technologies, Melbourne, Australia).

### 2.4. Methods

#### 2.4.1. Effect of Equilibrium pH

The sorption of Pb(II) and Ni(II) was carried out with batch experiments at 20 °C. A known amount of 0.1 g of CAM-D2EHPA was mixed (in polypropylene recipients) with synthetic solutions containing 10 mL of lead and nickel (50 mg L^−1^), at different pH conditions (controlled by the addition of HNO_3_ or NaOH, as required); the tubes were shaken at 180 rpm for 180 min, and then the solid phase was separated. The equilibrium pH (pHe) was measured by using a pH-meter from Jenco Electronics LTD (Missouri, TX, USA). The metal concentration was determined by an Agilent 4100 MP–AES Spectrometer at 405.78 and 352.45 nm wavelengths (for lead and nickel, respectively). The experiments were performed in triplicate and the standard deviation was estimated in the order of ±2%. The sorption efficiency (SE%) was assessed by Equation (1):(1)$$\mathrm{SE}\left(\%\right)=\frac{\left(C_{0}-C_{\mathrm{eq}}\right)}{C_{0}}\times 100$$
where *C*_0_ and *C*_eq_ are the initial and equilibrium metal concentrations in solution (mg L^−1^); respectively.

In order to evaluate the selectivity of the material under the binary system, a bimetallic solution was prepared at equimolar concentration (i.e., 0.24 mmol L^−1^ of Pb(II) and Ni(II)) and tested at different pHs with CAM-D2EHPA beads.

The separation factor (**β**) was calculated as the ratio of the amount of the sorbed metal, and the amount of metal present in the aqueous phase at equilibrium (Equation (2)).(2)$$\beta =\frac{D_{\mathrm{Pb}}}{D_{\mathrm{Ni}}}=\frac{{\left[\mathrm{Pb}\right]}_{\mathrm{sorbed}}/{\left[\mathrm{Pb}\right]}_{\mathrm{eq}}}{{\left[\mathrm{Ni}\right]}_{\mathrm{sorbed}}/{\left[\mathrm{Ni}\right]}_{\mathrm{eq}}}$$

#### 2.4.2. Sorption Isotherms

Metal sorption was carried out in reactors of polypropylene tubes at 20 °C. A known amount of air-dried sorbent (0.01 g) and 10 mL of aqueous metal solution (i.e., sorbent dosage, SD, of 1 g L^−1^) were used at different concentrations (10–1000 mg L^−1^). The pH was adjusted to 4, and after 3 h of agitation, the final pH was measured. The initial and equilibrium metal concentration were analyzed by 4100 MP–AES equipment. The amount of the metal ions sorbed onto the CAM-D2EHPA was calculated as follows Equation (3):(3)$$q=\frac{\left(C_{0}-C_{\mathrm{eq}}\right)V}{W}$$
where *q* is the sorption capacity of the material (mg g^−1^), *C*_0_ is the initial metal concentration in the solution (mg L^−1^), *C*_eq_ is the equilibrium concentration of the metal in the solution (mg L^−1^), *V* is the volume of the solution (L), and W is the mass of the sorbent (g).

Several sorption models are commonly used for describing the experimental data (such as Langmuir, Freundlich, and Sips equations) [16]. The Langmuir model is associated with homogeneous monolayer adsorption of the sorbate molecules onto the surface of sorbent [27]. The non-linear form of this isotherm is given by Equation (4):(4)$$q=\frac{q_{\max}bC_{\mathrm{eq}}}{1+bC_{\mathrm{eq}}}$$
where *q* is the amount of sorbed metal per gram of sorbent at equilibrium (mg g^−1^), *q*_max_ is the theoretical monolayer saturation capacity of the sorbent (mg g^−1^), the constant *b* (L mg^−1^) is the Langmuir equilibrium constant, and *C*_eq_ is the equilibrium concentration of the solution (mg L^−1^).

The non-linear form of the Freundlich equation is expressed as follows:(5)q=KFCeq1n
where *q* is the amount of sorbed metal per gram of sorbent at equilibrium (mg g^−1^), *K*_F_ and n are the Freundlich adsorption constants, indicative of the relative capacity and the adsorption intensity, and *C*_eq_ is the equilibrium concentration of the solution (mg L^−1^).

The Sips model is a combination of the Langmuir and Freundlich equations, and the non-linear form is represented as follows:(6)q=qmaxksCeq1ns1+ksCeq1ns
where *K*_S_ (L mg^−1^) is the affinity constant and *n*_S_ describes the surface heterogeneity of the materials. When 1/*n*_S_ close to 1, the Sips model reverts to the Langmuir equation (homogeneous surface); on the contrary, when the value is close to zero, the surface is generally supposed to be heterogeneous.

#### 2.4.3. Effect of Contact Time

The sorption kinetics experiments were carried out by adding 0.1 g of CAM-D2EHPA beads to 10 ml of metal solution (50 mg L^−1^) under continuous stirring (at 180 rpm), the initial pH values of aqueous solutions were adjusted to pH 4 using 0.1 M HNO_3_ solution. The final pH was measured after collecting the sample at different contact times; then, the metal solution was analyzed with the Agilent 4100 MP–AES equipment. The pseudo-first order and pseudo-second order rate equations (PFORE and PSORE, respectively) were used for fitting the experimental data.

Pseudo-first order rate Equation (PFORE):(7)$$\frac{dq_{t}}{dt}=K_{1}\left(q_{\mathrm{eq}}-q_{t}\right)$$

Pseudo-second order rate Equation (PSORE):(8)$$\frac{dq_{t}}{{\left(q_{\mathrm{eq}}-q_{t}\right)}^{2}}=K_{2}dt$$
where *q*_eq_ is the equilibrium sorption capacity (mg g^−1^), q_t_ is the sorption capacity (mg g^−1^) at any time *t* (min) and *k*_2_ is the pseudo-second order rate constant (g mg^−1^ min^−1^). The parameters *k*_1_ and *k*_2_ are pseudo-constants depending on the experimental conditions.

#### 2.4.4. Sorption–Desorption Experiments

The elution of metal ions from CAM-D2EHPA was performed with HNO_3_ solution (0.5 M) at room temperature. Between the sorption–desorption cycles the sorbent beads were washed with distilled water to remove the excess of eluent on the surface. The equilibrium pH of the sorption step was set as pH 2.5. The metal concentrations were measured after each sorption and desorption experiments. The desorption efficiency (DE%) of the cycles was calculated by the mass balance: (Desorbed mass/sorbed mass)*100.

#### 2.4.5. Dynamic System

A Fixed bed system was used for evaluating the sorption efficiency and the selectivity of CAM-D2EHPA towards lead and nickel ions. The continuous sorption–desorption experiments were performed in a glass column of 0.5 cm (internal diameter, i.d.) and 20 cm length. The CAM-D2EHPA beads were carefully packed in the column (0.52 g in dry weight, d.w.). A known volume of distilled water (25 mL) was passed through the fixed-bed before running the experiment, to improve the arrangement of the beads and to avoid the ‘side-wall’ effect. The distilled water was evacuated and then, the binary solution (at equimolar concentration of lead and nickel, 0.24 mmol L^−1^) was pumped from the bottom to the top of the column (up-flow) at room temperature using a peristaltic pump (with a constant flow rate of 0.13 mL min^−1^).

The tests were carried out at pH 2.2 in order to avoid metal precipitation and to compare with the results obtained in the batch experiments. After exhaustion, the sorbed metals were desorbed with dilute HNO_3_ solution (0.5 M) using the same flow-rate of the sorption step (0.13 mL min^−1^). Samples of 5 mL of effluent were periodically collected for analysis during the sorption and desorption steps using an automatic fraction collector (Gilson FC-203 B, Dunstable, UK). The breakthrough and the exhaustion points were set as *C*_t_/*C*_0_ = 0.05 and *C*_t_/*C*_0_ = 0.95, respectively. The experimental sorption capacity can be obtained from Equation (9):(9)$$q_{\exp}=\int_{0}^{V_{\mathrm{total}}}\frac{\left(C_{0}-C_{t}\right)}{m}dV$$

The Thomas model equation was used to describe the theoretical performance of the columns due to its simplicity and adequate accuracy in predicting breakthrough curves. The model can be represented by Equation (10) [16]:(10)$$\frac{C_{t}}{C_{0}}=\frac{1}{1+\exp\left[K_{T}\left(q_{T}m-C_{0}V\right)/Q\right]}$$
where *K*_T_ is the Thomas rate constant (L h^−1^ mg^−1^), m is the mass of sorbent (g), *Q* is the volumetric flow rate (L h^−1^), *V* is the volume of the solution into the column (L), *C*_t_ is the metal concentration (mg L^−1^), and *q*_T_ is the Thomas sorption capacity (mg g^−1^). The constants of the non-linearized form were obtained by origin 9.0 software (OriginLab Inc., Northampton, MA, USA, 2012).

## 3. Results and Discussion

### 3.1. Characterization of the Materials

#### 3.1.1. FTIR Analyses

FTIR analyses were used in order to identify the main bands corresponding to the functional groups of the sorbents (Figure 1). The characteristic FTIR bands of calcium alginate are shown in Figure 1a. Two specific bands at 1616 and 1419 cm^−1^ can be attributed to carboxyl groups (COO–) [25]. The bands at 3450, 2918, 1085, and 1033 cm^−1^ are attributed to –OH hydroxyl groups (stretching), –CH_2_ (stretching), C–O–H (stretching), and C–O–C (stretching) vibrations, respectively [29].

Figure 1b shows the IR spectrum of HM-D2EHPA. The shoulder band at 1076 cm^−1^ is assigned to the Si–O stretching vibration; the bands at 705 and 785 cm^−1^ are attributed to the stretching and asymmetric bend vibrations of Si–O–Si, respectively [38]. The bands between 3443 and 3590 cm^−1^ correspond to the hydroxyl groups of H-magadiite [24]. The band appearing at 2960 cm^−1^ was attributed to the C–H stretching of D2EHPA extractant. The results are in agreement with Darvishi et al. [39], the main characteristic bands of D2EHPA (i.e., P=O and C–H bending) occur at 1206, and 1466 cm^−1^, respectively. The spectrum of CAM-D2EHPA (Figure 1c) shows the characteristic bands of both CA and HM-D2EHPA materials, this confirms the total immobilization of the synthesized powder into the alginate beads.

#### 3.1.2. SEM–EDX Analyses

Air-dried beads of CAM-D2EHPA were studied with scanning electron microscopy (SEM), to examine the surface morphology. Figure 2a shows the spherical shape of the sorbent; the surface is rough and the HM-D2EHPA particles seem to be homogeneously dispersed over the entire topography of the sorbent. The immobilization with alginate allows a constant shape to be maintained (with an average diameter of 1.8 mm), which is more operationally useful than the irregular powder particles, for evaluating in columns systems.

EDX-analyses of the CAM-D2EHPA (Figure 2b) confirm the presence of the main elements of alginate such as carbon, oxygen, calcium, chloride, and sodium as expected [16]. Additionally, the presence of silicon from H-magadiite (Si–O bond), and phosphorous from D2EHPA (P–OH bond) was detected in the whole volume of the material (the beads were dissected and the EDX-analyses show the presence of the same elements in the different surface zones). This is evidence of the good distribution of the particles in the beads.

### 3.2. Effect of pH

The pH of the solution plays an important role in the removal of cations from an aqueous medium. The pH may affect the speciation of the metals through formation of complex molecules [40], and the protonation of carboxylic groups present in the alginate chains (the mannuronic and guluronic blocks; i.e., M and G-blocks) [41]. Figure 3 shows the influence of pH on the sorption efficiency of lead and nickel from aqueous solutions using CAM-D2EHPA material as sorbent. In order to avoid a precipitation phenomenon, the initial pH was varied within the pH range of 1.0–5.5 at dilute metal concentration (50 mg L^−1^). According to the speciation diagram (Appendix A), the lead and nickel hydroxides precipitate at pH ≥ 5.5 and pH ≥ 6.5, respectively.

The lower sorption efficiency (below 50%) obtained at pH ≤ 2.0 is partially due to the strong competition between H^+^ and M^2+^ species, which can also be related to the diffusion into actives sites of the sorbent [41]. In the same range of pH, the sorption efficiency of lead is better in comparison with nickel, this is the first indicative of the higher affinity of CAM-D2EHPA toward Pb(II) ions. The metal removal is favored at pH above 4, the sorption performance increases up to 95% for lead and 80% for nickel (Figure 3a). This can be explained by two possible mechanisms participating: (i) metal binding through carboxylic groups of the alginate, which was particularly evidenced by the competition between the metals and the protons of the solution (Figure 3b), a buffer-effect is produced between pH 3.5 and 5.0; (ii) Ion-exchange interactions between H-magadiite and phosphoric groups (P–OH) of D2EHPA with the metal species.

The points of zero charge (so-called pH_PZC_) of the sorbent and their main components were determined by the drift method [26]. Appendix A shows that the introduction of H-magadiite impregnated with D2EHPA into the alginate matrix, contributes to reduce the pH_PZC_ values from 6.2 to 5.9. This finding improves the metal binding of alginates and enhances the electrostatic sorbent/sorbate interactions. At pH < pH_PZC_, the surface of the sorbent is positively charged, this justifies the low removal of lead and nickel at acidic conditions. By increasing the pH (pH > pH_PZC_), the surfaces become more negative, which promotes the attraction between the sorbent and the metal cations [42].

The selectivity of the sorbent was studied by using a bi-metallic solution of lead and nickel at equimolar concentration (0.24 mmol L^−1^) and at different pH conditions. The separation factor β(D_Pb_/D_Ni_) values for Pb(II) and Ni(II) with equilibrium pH are given in Table 1. It is noteworthy that the β values are significantly greater than 1, which confirm the high affinity toward Pb(II) ions [33].

The selectivity of the sorbent increases with a decrease in the pH values; this can be attributed to the effect of the introduction of impregnated H-magadiite with D2EHPA into the sorbent, which was demonstrated previously to be strongly selective toward lead ions [23]. The contribution of carboxylic groups to the metal removal at pH < 3.4 is negligible since H^+^ ions of the medium tend to compete for the active sites. Haug et al. [43] reported pKa values of COO^−^ groups in mannuronic and guluronic acids as 3.38 and 3.65, respectively; it means that at pH < 3.4 the carboxylic groups are particularly protonated and the interactions with the metal species are weak, this in agreement with Demey et al. [16]. According to Papageorgiou et al. [44] and Wang et al. [33], the selectivity of the metals also depends on the valence orbital energy (X_m_^2^r: 531 for lead and 392 for nickel), and according to Bohli et al. [45] it can additionally be correlated to the hydrolysis constants (pKh: 7.80 for lead and 9.40 for nickel).

### 3.3. Sorption Isotherms

The sorption isotherms of Pb(II) and Ni(II) (Figure 4) were studied to interpret the distribution of the solute between the liquid and the solid phase at equilibrium. The maximum loading capacity of metal on 1 g of CAM-D2EHPA (air-dried beads) was determined; the initial concentration of the solutions was varied from 10 to 1000 mg L^−1^ for each metal, at pH 4 and at room temperature. Figure 4 shows that an increase in the metal concentration results in an increase in the sorption capacity until a saturation plateau is achieved; the maximum sorption capacity of lead and nickel was 197 mg g^−1^ and 44 mg g^−1^, respectively. At dilute metal concentration (i.e., <10 mg L^−1^) the sorbent recovered more than 94% and 65% of Pb(II) and Ni(II), respectively. This gives evidence for the high performance and the feasibility of CAM-D2EHPA for separating trace metals from dilute wastewaters.

The fitting values of the Langmuir, Freundlich, and Sips models are given in Table 2. The experimental data is better described using the Langmuir and Sips equations (by comparison of the determination coefficients, *r*^2^). The values of 1/*n*_S_ (Sips model) for lead and nickel are closer to 1, indicating a homogeneous distribution of the sorbent sites into the sorbent, and the Sips model reverts to the Langmuir equation [46].

The affinity parameter *K*s (Table 2) of lead is a little higher compared to the *K*s values of nickel, these differences are attributed to the type of reactive groups present in the sorbent, and to the physical–chemical characteristics of the metal species. Similar order of affinity (Pb(II) >Ni(II)) was obtained by Demey et al. [16] and Bhattacharyya et al. [47]; the new CAM-D2EHPA material possesses more affinity towards Pb(II) and Ni(II)) compared to the calcium alginate beads (Appendix A). It is directly associated to the presence of D2EHPA in the composite.

In addition, Table 3 shows a comparison of the maximum sorption capacities of the materials reported in the literature; e.g., CAM-D2EHPA has a greater nickel sorption capacity compared to the impregnated solids studied by Gonzalez et al [48] (XAD-2-Cyanex 272). The removal of lead is also comparable with the results obtained by Pereira et al. [17], Bhattacharyya et al. [47], Asuquo et al. [49], and Puppa et al. [50]. CAM-D2EHPA has two important advantages: (i) it is easy to manufacture, which is interesting for industrial scale production, and (ii) it has a high sorption uptake. Abu Al-Rub et al. [51] obtained a poor adsorption of nickel with calcium alginate beads (25.60 mg g^−1^). The introduction of powder impregnated with D2EHPA into the beads may increase the affinity and the sorption metal uptake.

### 3.4. Sorption Kinetics

The kinetics of sorption is important for selecting the best sorbent, as a function of the required contact time for achieving equilibrium. Figure 5 plots the kinetic profiles of lead and nickel using CAM-D2EHPA as sorbent. The removal of Pb(II) and Ni(II) have a similar trend and three pseudo steps can be characterized: (i) a first step that lasts about 20 min, and the sorption uptake is faster (the efficiency increased from 36% to 72% for Pb(II) and from 45% to 75% for nickel, respectively); (ii) a second step, in which the sorption uptake is slower, and lasts about 20–180 min (the yield increased from 73% to 92% for Pb(II) and from 76% to 79.6% for nickel); (iii) a third step that takes 3 h approximately, and the uptake is very slow (the sorption yield increased to 97% for lead and to 85.7% for nickel).

The calculated values of PFORE and PSORE are given in Table 4, the sorption kinetic profiles of Pb(II) and Ni(II) can be described by both pseudo-first and pseudo-second order models. Although PSORE was originally used for describing chemisorption processes [57]; the good fitting of the experimental data does not necessarily mean that the principles of the models are demonstrated; nevertheless it is useful for interpreting the involved mechanisms.

### 3.5. Sorption–Desorption Cycles

A good sorbent, especially for large-scale application, must be recyclable in order to be competitive [13]. The reusability of CAM-D2EHPA was demonstrated by ten consecutives sorption–desorption cycles using single solutions of lead and nickel at dilute concentration (50 mg L^−1^).

The results reported in Table 5 (and Appendix A) confirm the high performance of the sorbent toward the target metals; sorption efficiencies in the order of 98% for lead and 80 % for nickel were obtained. It was demonstrated that HNO_3_ (0.5 M) is a good eluent for metal recovery from the loaded beads; the obtained desorption rate was in the order of 88%–96% for lead and 93%–99% for nickel. The mechanical stability of the sorbent was verified and monitored during the cycles; CAM-D2EHPA conserves the characteristic spherical shape. The beads tend to swell and after ten elution cycles, the average diameter (Ø) was found to be 2 mm; these are similar to the original hydrated beads (before air-drying), which have an average diameter of 1.8 mm.

On the contrary, the physical evaluations carried out with the raw calcium alginate beads (CA) showed that after three cycles, the CA beads tend to swell very rapidly as a result of the fragile stability in the presence of sodium salts (Appendix A); these are not stable in effluents containing Na^+^ (even at low ionic-strength: 0.005 M NaNO_3_). After ten cycles, the average diameter of CA was found to be two times greater than CAM-D2EHPA (i.e., 3.9 mm). The introduction of HM-magadiite/D2EHPA into the alginate matrix improves the mechanical strength of the resulting CAM-D2EHPA material, making this material appropriate for industrial applications (this will be the scope of a future work).

### 3.6. Dynamic System

Columns experiments are very useful for evaluating CAM-D2EHPA beads in similar conditions to the real wastewater treatment application. It is noteworthy that most of the sorption studies present in the literature have been reported using a batch system; the dynamic system experiments are very important since they allow the design and the scaling-up of the sorption technique to be improved.

The simplicity of using a fixed amount of packed sorbent for treating large volumes of contaminated effluents is the main advantage of the column configuration; it involves a better engineering conception of the material, and a better knowledge of the operating conditions. Additionally, the column configuration allows several steps of the batch system to be reduced (such as agitation of the batch reactor, separation of the solid sorbent from the aqueous phase, etc.). Figure 6 shows the typical S-shape of the curves by using a simulated effluent of lead and nickel (at equimolar inlet concentration of 0.24 mmol L^−1^ and pH 2.2). A dilute concentration of the solution was chosen in order to be close to industrial wastewater [16].

The breakthrough curves (Figure 6) have three relevant features: (i) the first step of the curves is more prolonged for Pb(II) than for Ni(II), indicating the special preference of the sorbent toward lead ions; (ii) the bed-volumes (BV) required for achieving the breaking point (*q*_BP_) are 10 times greater for Pb(II) than for Ni(II), i.e., 59.1 bed-volumes for Pb(II) and 5.9 bed-volumes for Ni(II) (it means that 5% of the feed concentration is leached after 5.9 bed-volume; Table 6); (iii) Nickel in the presence of lead is rapidly eluted from the column. Lead ions compete with nickel for active sites in the sorbent, thus nickel overshoot is produced; the sorbent has a limited sorption capacity and the species with lower affinity are pushed-off and displaced by the species of higher affinity. This phenomenon is in agreement with Sag and Kutsal [58].

Table 6 reports the sorption capacity obtained by the fixed-bed CAM-D2EHPA at the breaking and exhaustion points (*q*_BP_ and *q*_exp_, respectively); for lead, 22.3 mg g^−1^ at the breaking point and 61.5 mg g^−1^ at the exhaustion point were obtained. The sorption capacity corresponds well to the preliminary results obtained in the binary system (Table 1), but it is almost three times lower of that obtained in the batch system at single conditions (Table 2). The difference in sorption capacity values (between single and binary system) could be due mainly to the poor mass transfer in the column, which is associated with the resistance to film diffusion (particularly active in dilute concentrations) [59]. The mass transfer is improved on reducing the flow rate [13]. As expected, the removal of nickel was low (5.7 mg g^−1^); the sorbent is very selective toward lead ions at the chosen operating conditions. This agrees with the results of Table 1.

It is noteworthy that sorption unities are commonly used as a “polishing” step for treating volumes of effluents containing traces of pollutants., and these are usually installed downstream of the primary techniques (e.g., coagulation–flocculation) in industrial water treatment applications [13]. Thus, the recycling of the fixed-bed is important to achieve a high performance and reduce the operational costs of the process. The elution of the column was performed with dilute HNO_3_ solution (0.5 M). As expected, the sorbed lead was entirely recovered (Figure 7): the elution efficiency was found to be 99% for lead and 21.5% for nickel. The volume required for the fixed-bed regeneration was estimated as 0.05 L. This makes the large-scale utilization of CAM-D2EHPA for the selective sorption of lead from industrial effluents promising and feasible.

## 4. Conclusions

In the present work, the manufactured CAM-D2EHPA material was especially conceived for the selective removal of lead and nickel ions from nitrate solutions; the introduction of HM-D2EHPA in the calcium alginate gel involves the addition of new functional groups (e.g., phosphoryl groups), improving the sorption uptake for the target metal ions at the same operating conditions. The sorption of Pb(II) and Ni(II) ions by CAM-D2EHPA beads was strongly influenced by the pH solution; an increase in pH, increases the sorption efficiency. The beads have a high selectivity toward lead in the presence of nickel ions over a wide pH range (at equimolar concentrations); this selectivity toward Pb(II) was attributed to the presence of extractant (D2EHPA) in the hybrid sorbent. The sorption isotherm data were better fitted with the Langmuir equation, the maximum sorption capacity was found to be 197 and 44 mg g^−1^ for lead and nickel, respectively. A contact time of about 180–240 min was required to achieve equilibrium at pH 4; the sorption kinetic data followed a pseudo-second order rate equation (PSORE).

The regeneration of CAM-D2EHPA sorbent was examined with 10 consecutive sorption−desorption cycles using a dilute solution of HNO_3_ (0.5 M); sorption efficiencies in the order of 98% and 80% for lead and nickel respectively were obtained. The possibility of using CAM-D2EHPA beads in a fixed-bed column was demonstrated with a simulated effluent of lead and nickel (in a binary system). The results were in agreement with the batch system, and the experimental data were accurately described with the Thomas model. The main advantages of using CAM-D2EHPA are the very simple manufacturing procedure, low-cost, high mechanical strength, added to high capacity for sorption of heavy metals compared to the hybrid materials reported in the literature.

## Figures and Tables

**Figure 1 polymers-11-00340-f001:**
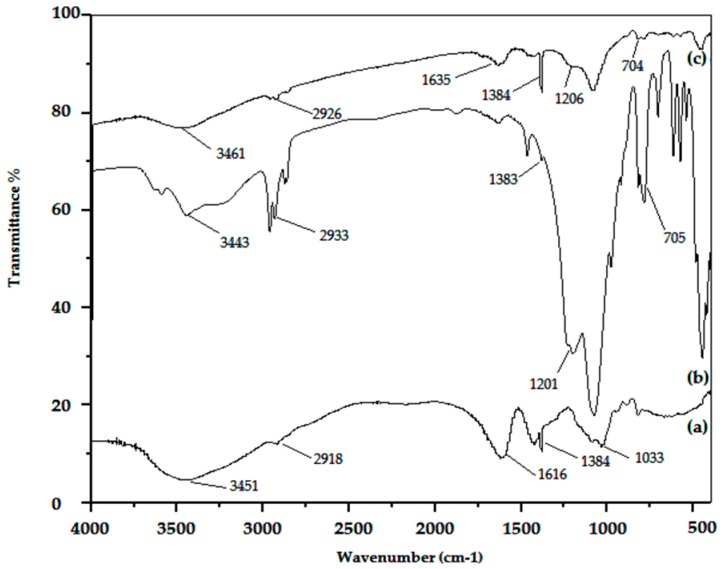
FTIR analyses of the sorbent material. (**a**) CA; (**b**) HM-D2EHPA; (**c**) CAM-D2EHPA.

**Figure 2 polymers-11-00340-f002:**
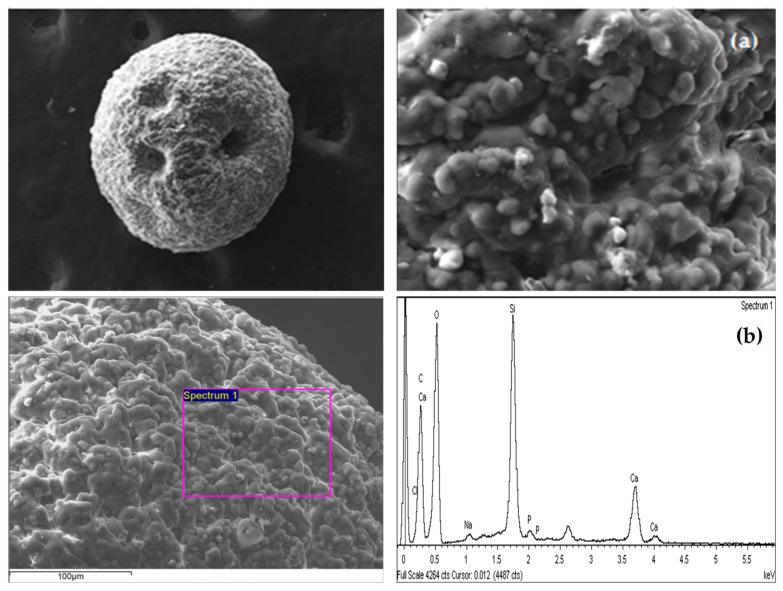
Scanning Electron Microscopy (SEM) images of the CAM-D2EHPA material: (**a**) Topography of the sorbent; (**b**) Energy Dispersive X-ray (EDX) analysis on the surface area of the solid.

**Figure 3 polymers-11-00340-f003:**
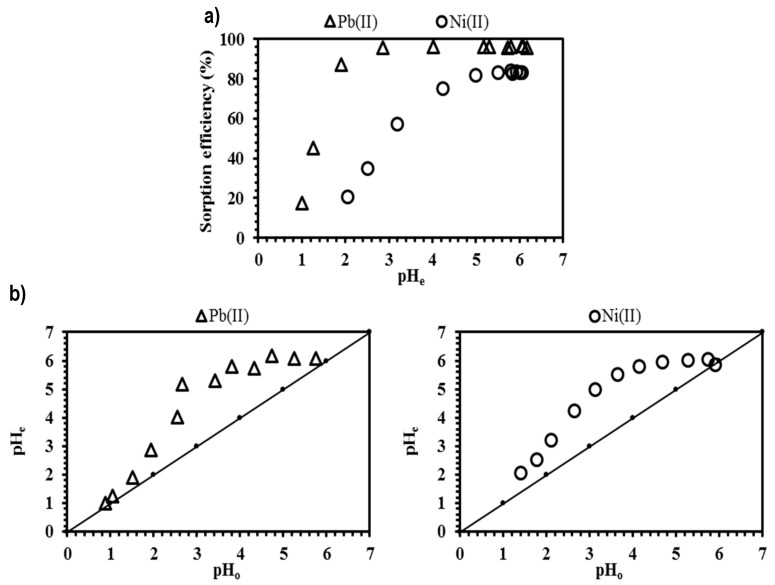
Influence of pH on lead and nickel removal: (**a**) Sorption efficiency; (**b**) Variation in pH using CAM-D2EHPA as sorbent (*T*: 20 °C; *V*: 0.01 L; *m*: 0.1 g; agitation speed: 180 rpm; contact time: 3 h; *C*_0_: 50 mg L^−1^; 0.005 M NaNO_3_).

**Figure 4 polymers-11-00340-f004:**
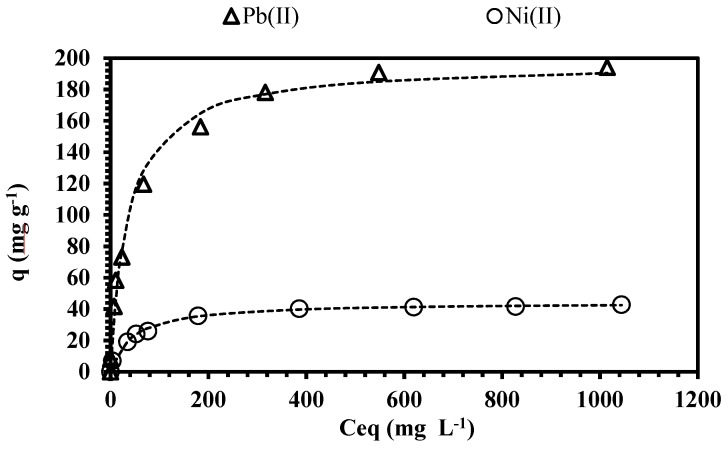
Isotherm plots for lead and nickel removal. (Dashed line: Langmuir model; *T*: 20 °C; *V*: 0.01 L; *m*: 0.01 g; agitation speed: 180 rpm; contact time: 3 h; pH: 4; *C*_0_: 10–1000 mg L^−1^; 0.005 M NaNO_3_).

**Figure 5 polymers-11-00340-f005:**
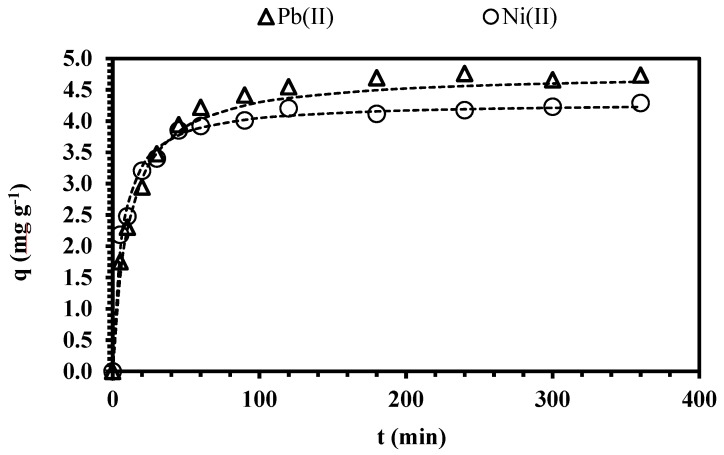
Effect of contact time on CAM-D2EHPA dried beads. (Dashed line: Pseudo-second order model (PSORE) model; *T*: 20 °C; *V*: 0.01 L; *m*: 0.1 g; agitation speed: 180 rpm; contact time: 3 h; pH: 4; *C*_0_: 50 mg L^−1^, 0.005 M NaNO_3_).

**Figure 6 polymers-11-00340-f006:**
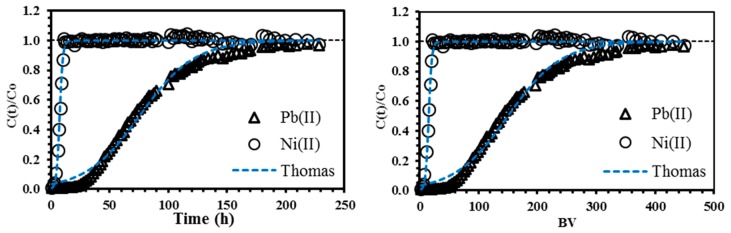
Breakthrough curves for a binary system of Pb(II) and Ni(II) using hydrated CAM-D2EHPA as sorbent (dashed line: Thomas model; *T*: 20 °C; internal diameter of the column, Ø: 0.5 cm; column height: 20 cm; flow rate: 0.13 mL min^−1^; pH: 2.2; *C*_0_: 0.24 mmol L^−1^).

**Figure 7 polymers-11-00340-f007:**
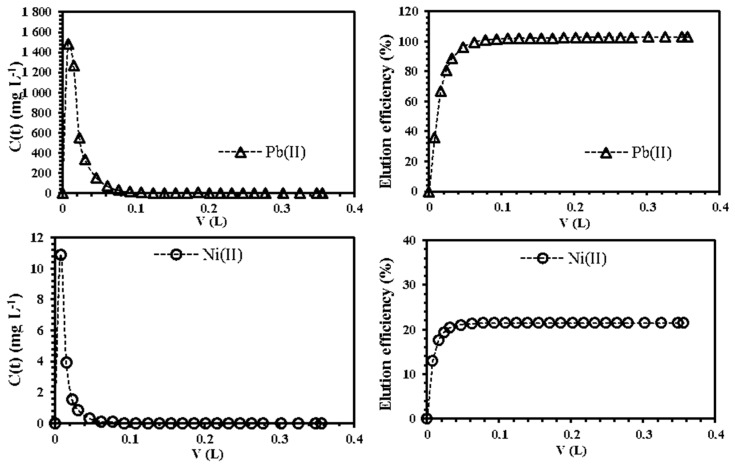
Simultaneous elution of Pb(II) and Ni(II) from the fixed-bed column by using HNO_3_ (0.5 M) as eluent (**Left side**: Recovered metal concentration as a function of the volume (L), dashed line: desorption trend; **right side**: Elution efficiency as a function of the volume (L); *T*: 20 °C; internal diameter of the column, Ø: 0.5 cm; column height: 20 cm; flow rate: 0.13 mL min^−1^).

**Table 1 polymers-11-00340-t001:** Selectivity of Pb(II) and Ni(II) using CAM-D2EHPA material as sorbent.

Initial pH	Equilibrium pH	SE _Pb(II)_ (%)	SE _Ni(II)_ (%)	*D* _Pb_	*D* _Ni_	$\beta =D_{Pb}/D_{Ni}$
1.27	1.64	87.2	0.6	6.8	0.01	1136.5
1.44	2.10	95.8	11.2	22.5	0.13	179.4
1.58	2.83	97.9	42.6	48.6	0.74	65.6
1.87	2.92	98.4	47.2	59.5	0.90	66.7
2.05	3.34	99.1	62.5	104.4	1.66	62.7

**Table 2 polymers-11-00340-t002:** Langmuir, Freundlich, and Sips constants of CAM-D2EHPA sorbent.

Experimental		Langmuir			Freundlich			Sips			
Metal	*q*_exp_ (mg g^−1^)	*q*_max_ (mg g^−1^)	*k*_L_ (L mg^−1^)	*r* ^2^	*k*_F_ (mg^1−1/*n*^ g^−1^ L^1/*n*^)	*n* _F_	*r* ^2^	*q*_max_ (mg g^−1^)	*k*_S_ (L mg^−1^)	*n* _S_	*r* ^2^
Pb(II)	197.17	197.26	0.027	0.990	33.55	3.68	0.955	221.68	0.054	1.36	0.998
Ni(II)	42.84	44.36	0.022	0.992	9.20	4.31	0.957	47.73	0.050	1.32	0.996

**Table 3 polymers-11-00340-t003:** Metal removal by different sorbents in the literature.

Sorbents	Experimental Conditions	Sorption Capacity (mg g^−1^) Pb(II)	Sorption Capacity (mg g^−1^) Ni(II)	References
Silica gel (SG) Modified silica gel (S2A)	pH = 2.0	0.72 -	0.34 0.95	[18]
Kaolinite Acid-activated kaolinite	pH = 7.0	11.10 12.10	10.40 11.90	[47]
Mesoporous activated carbon adsorbent	pH = 7.0	20.32	-	[49]
EIRs	pH = 1.0	80.00	-	[52]
4-amino-2-mercaptopyrimidine modified silica gel	pH = 3.0	80.20	-	[17]
Fe_3_O_4_-SO_3_H MNPs	pH = 7.0	108.93	-	[53]
ChiFer(III)	pH = 4.5	116.03	-	[54]
Amorphous manganese oxide (AMO)	pH = 4.0 pH = 5.5	124.32 125.15	- -	[50]
CAM-D2EHPA	pH = 4.0	197.26	44.36	[This work]
AF-PEI A-PEI	pH = 4.0	225.85 229.99	60.45 48.71	[16]
XAD-2-Cyanex 272 XAD-2-Cyanex 302	pH = 2.0	-	6.49 3.65	[48]
Free dead algal cells Blank alginate beads Immobilized dead algal cells	pH = 5.0	- - -	13.90 25.60 31.30	[51]
Activated carbon Irradiation grafted Activated carbon	pH = 7.0	- -	44.10 55.70	[55]
Raw poplar (trunk) Torrefied poplar (250 °C, 75 min) Torrefied poplar (280 °C, 60 min)	pH = 4.0 pH = 4.0 pH = 4.0	28.50 27.55 30.89	- - -	[56]

**Table 4 polymers-11-00340-t004:** Kinetic parameters of CAM-D2EHPA sorbent.

Experimental		Pseudo-First Order Model (PFORE)			Pseudo-Second Order Model (PSORE)		
Metal	*q*_exp_ (mg g^−1^)	*k*_1_ (min^−1^)	*q*_1_ (mg g^−1^)	*r* ^2^	*k*_2_ (g mg^−1^ min^−1^)	*q*_1_ (mg g^−1^)	*r* ^2^
Pb(II)	4.71	0.057	4.57	0.992	0.019	4.88	0.994
Ni(II)	4.22	0.097	4.07	0.954	0.037	4.30	0.992

**Table 5 polymers-11-00340-t005:** Sorption and desorption cycles using CAM-D2EHPA sorbent.

Cycles	Pb(II)			Ni(II)		
Number of Cycles	pHe	Sorption (%)	Desorption (%)	pHe	Sorption (%)	Desorption (%)
1	2.5	97.8	94.0	2.5	83.2	99.1
2	2.6	97.8	92.5	2.6	74.8	99.5
3	2.5	97.6	88.8	2.5	74.8	99.6
4	2.6	97.7	97.5	2.7	80.7	93.7
5	2.6	97.9	89.3	2.6	76.5	92.6
6	2.6	97.7	94.1	2.6	74.7	95.0
7	2.6	97.7	89.0	2.6	76.3	93.8
8	2.6	98.3	87.0	2.6	74.9	96.5
9	2.6	98.3	81.4	2.6	77.2	94.7
10	2.6	98.5	88.0	2.6	77.2	95.7

**Table 6 polymers-11-00340-t006:** Sorption parameters for continuous system using CAM-D2EHPA material as sorbent.

Experimental				Thomas Parameters		
Sorbate	*q*_exp_ (mg∙g^−1^)	*q*_BP_ (mg∙g^−1^)	BV_BP_	*q*_T_ (mg∙g^−1^)	*K*_T_ (L∙h^−1^∙mg^−1^)	*r* ^2^
Pb(II)	61.50	22.3	59.13	56.71	8.62 × 10^−4^	0.986
Ni(II)	5.51	0.58	5.91	5.69	1.44 × 10^−2^	0.992

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
