# Peer review of "Sorption and Desorption Studies of Pb(II) and Ni(II) from Aqueous Solutions by a New Composite Based on Alginate and Magadiite Materials"

_polymers, 2019, doi:10.3390/polym11020340_

Round 1

Reviewer 1 Report

1. Figure 1 should be informed in clear, concise and high resolution format.

2. Figure 2 should be informed in clear, concise and high resolution format.

3. what are the significance to find results in this paper in Table 1?

4. what does senses of "The results reported in Table 5 (and Figure S3)"?

5. pls check the unity index of ISU in line 441.

6. Figure 6 should be informed in clear, concise and high resolution format. and pls show the curve as a statedy state level curve?

7. please revise the abstract more precisely.

8. Author may include the following references.

[1].          Mingliang Ge, Wei Tang, Mingyi Du, Guodong liang, Guoqing Hu, Jahangir Alam S.M.*, Research on 5-Fluorouracil as a drug carrier materials and its in vitro release properties on organic modified magadiite, European Journal of Pharmaceutical Sciences, 130(2019), 44-53. (SCI indexed)

[2].          Ge, M.; Wang, X.; Du, M.; Liang, G.; Hu, G.; S.M., J.A. (Jahangir Alam S.M.*), Effects on the Mechanical Properties of Nacre-Like Bio-Hybrid Membranes with Inter-Penetrating Petal Structure Based on Magadiite. Materials 2019, 12, 173.

[3].          Ge M.; Wang, X.; Du, M.; Liang, G.; Hu, G.; S.M., J.A (Jahangir Alam S.M.*). Adsorption Analyses of Phenol from Aqueous Solutions Using Magadiite Modified with Organo-Functional Groups: Kinetic and Equilibrium Studies. Materials 2019, 12, 96.

[4].          Mingliang Ge, Luoxiang Cao, Mingyi Du, Jahangir Alam S.M.*, Adsorptive characterization of a pure magadiite and an organic modified magadiite on removal of methylene blue from related aqueous solution, Materials Chemistry and Physics, 2018, 217, 533-540. (SCI).

[5].          Mingliang Ge, Luoxiang Cao, Mingyi Du, Guoqing Hu, Jahangir Alam S.M.*, Competitive adsorption analyses of a pure magadiite and a New silylated magadiite on methylene blue and phenol from related aqueous Solution, Materials Chemistry and Physics, 2018, 217, 393-402. (SCI).

[6]. Mingliang GeMingyi DuLuoyun ZhengBingying WangXiangyang ZhouZhixin JiaGuoqing HuS.M. Jahangir Alam*, A Maleic Anhydride Grafted Sugarcane Bagasse Adsorbent and Its Performance on the Removal of Methylene Blue from Related Wastewater, Materials Chemistry and Physics, May 2017, 192, 147-155. (SCI).

Author Response

Hary Demey

Universitat Politecnica de Catalunya

Chemical Engineering Department

Av. Diagonal, 647.

08028 Barcelona, Spain

Dr. Jun-Ichi Kadokawa,

Guest Editor of Polymers (MDPI Journal)

Dear Editor:

Please find enclosed the revised manuscript entitled Sorption and desorption studies of Pb(II) and Ni(II) from aqueous solutions by a new composite based on alginate and magadiite materials”  to Polymers (MDPI Journal). We confirm that this work has not been published, nor is it currently under consideration for publication elsewhere.

We would like to thank the referees for the careful review of the work and their useful suggestions for improving it. A thorough revision has been conducted following all their relevant considerations. This manuscript describes the manufacturing procedure of the new composite based on alginate and magadiite materials. It provides information about the immobilization of layered silicate magadiite (impregnated with D2EHPA) into calcium alginate beads. The environmentally-friendly CAM-D2EHPA material was tested for lead and nickel removal from aqueous solutions.

The CAM-D2EHPA beads were demonstrated to be promising for the selective separation of lead from synthetic effluents containing Pb(II) and Ni(II) as main pollutants (at equimolar concentration). The influences of the main parameters (i.e. pH, initial metal concentration, contact time, ionic-strength) in batch system were considered. The introduction of magadiite/D2EHPA into the alginate support, improves the metal selectivity toward Pb(II) ions.

The evaluations in the column configuration with a binary system Pb(II)-Ni(II) confirmed the reproducibility of treating effluents with a continuous process; this is particularly useful for industrial applications. The sorption-desorption cycles confirmed the easy recovery of the metals and the rapid regeneration of the sorbent with HNO3 (0.5 M) solutions was demonstrated. Also, the 10 sorption cycles allowed evaluating the mechanical stability of the CAM-D2EHPA beads; these were found to be physically stronger than the raw calcium alginate beads in presence of sodium ions. The composite CAM-D2EHPA can be used for industrial effluents, and the results obtained in this work can be extrapolated to different species of metals; this is a doorway for a new generation of green-sorbents.

As such, this paper could be of interest to a broad readership including those interested in polymer chemistry, polysaccharides applications, chemical engineering, material science and water treatment. We hope you will agree this manuscript; however, we are prepared to modify the paper according to reviewers’ comments.

Sincerely,

PhD. Hary Demey

Reviewer #1:

1. Figure 1 should be informed in clear, concise and high resolution format.

2. Figure 2 should be informed in clear, concise and high resolution format.

3. What is the significance to find results in this paper in Table 1?

4. What does senses of "The results reported in Table 5 (and Figure S3)"?

5. Please check the unity index of ISU in line 441.

6. Figure 6 should be informed in clear, concise and high resolution format. And please show the curve as a statedy state level curve?

7. Please revise the abstract more precisely.

8. Author may include the following references.

[1]. Mingliang Ge, Wei Tang, Mingyi Du, Guodong liang, Guoqing Hu,Jahangir Alam S.M.*, Research on 5-Fluorouracil as a drug carrier materials and its in vitro release properties on organic modified magadiite, European Journal of Pharmaceutical Sciences, 130(2019), 44-53. (SCI indexed)

[2]. Ge, M.; Wang, X.; Du, M.; Liang, G.; Hu, G.; S.M., J.A. (Jahangir Alam S.M.*), Effects on the Mechanical Properties of Nacre-Like Bio-Hybrid Membranes with Inter-Penetrating Petal Structure Based on Magadiite. Materials 2019, 12, 173.

[3]. Ge M.; Wang, X.; Du, M.; Liang, G.; Hu, G.; S.M., J.A (Jahangir Alam S.M.*). Adsorption Analyses of Phenol from Aqueous Solutions Using Magadiite Modified with Organo-Functional Groups: Kinetic and Equilibrium Studies. Materials 2019, 12, 96.

[4]. Mingliang Ge, Luoxiang Cao, Mingyi Du, Jahangir Alam S.M.*, Adsorptive characterization of a pure magadiite and an organic modified magadiite on removal of methylene blue from related aqueous solution, Materials Chemistry and Physics, 2018, 217, 533-540. (SCI).

[5]. Mingliang Ge, Luoxiang Cao, Mingyi Du, Guoqing Hu, Jahangir Alam S.M.*, Competitive adsorption analyses of a pure magadiite and a New silylated magadiite on methylene blue and phenol from related aqueous Solution, Materials Chemistry and Physics, 2018, 217, 393-402. (SCI).

[6]. Mingliang Ge, Mingyi Du, Luoyun Zheng, Bingying Wang, Xiangyang Zhou, Zhixin Jia, Guoqing Hu, S.M. Jahangir Alam*, A Maleic Anhydride Grafted Sugarcane Bagasse Adsorbent and Its Performance on the Removal of Methylene Blue from Related Wastewater, Materials Chemistry and Physics, May 2017, 192, 147-155. (SCI).

Authors:

We are very grateful to reviewer #1 for the valuable comments. The modifications were highlighted (in red color) in the whole text.

1. Thank you for your recommendation.

Figure 1 has been modified with a higher resolution format. The resolution was increased from 72 dpi to 300 dpi

Figure 1. FTIR analyses of the sorbent material. (a) CA; (b) HM-D2EHPA; (c) CAM-D2EHPA

2. Figure 2 has been modified with a higher resolution format and now it is more clear and concise. The resolution was increased from 72 dpi to 300 dpi.

Figure 2. Scanning Electron Microscopy (SEM) images of the CAM-D2EHPA material: (a) Topography of the sorbent; (b) Energy Dispersive X-ray (EDX) analysis on the surface area of the solid.

3. The authors confirm that the importance of the reported results in Table 1, is the demonstration of the pH-dependence of the process. It was found that at high pH conditions, the sorption efficiency is better. Nevertheless, at equilibrium pH 1.6, the separation factor (DPb/DNi) of the binary system is optimum. It indicates that the sorbent is highly selective toward Pb(II) ions in presence of Ni(II) (at equimolar concentration); thus, the sorption of lead from aqueous effluents is more favorable.

The authors confirm that experiments were performed in triplicate and the standard deviation of the results was estimated in the order of + 2%. The t-student test was carried out and the difference is not significant.

The following sentence was added in the manuscript (page 4, line 182):

The experiments were performed in triplicate and the standard deviation of the results was estimated in the order of + 2%.

4. The authors understand the reviewer’s point of view. The obtained results in Table 5 are of great importance since allowed demonstrating the reuse possibility of the sorbent beads. Ten sorption-desorption cycles can be performed with CAM-D2EHPA material with no detrimental the sorption efficiency. The sorption efficiencies of the cycles could be of interest to readership, which could be reproduced in a major industrial scale.

5. The index in line 441 was verified and corrected.

6. The authors agree with the reviewer. The resolution of Figure 6 was increased.

The breakthrough curves are commonly presented as the variation of the metal concentration as a function of the operating time (i.e., C/C0 vs time); or also as the bed-volumes as a function of time. This allows a better understanding of the column operation and enhances the fitting of the models such as Thomas, and Bohart-Adams equations.

The breakthrough curves presentation is comparable with those reported by Demey et al. [13], and Demey et al. [16], in the sorption of heavy metals and neodymium, respectively:

[13] Demey, H.; Lapo, B.; Ruiz, M.; Fortuny, A.; Marchand, M.; Sastre, A.M. Neodymium recovery by chitosan/Iron(III) hydroxide [ChiFer(III)] sorbent material: Batch and Column systems, Polymers 2018, 10(2), 204: https://doi.org/10.3390/polym10020204.

[16] Demey, H.; Vincent, T.; Guibal, E.; A novel algal-based sorbent for heavy metal removal. Chem. Eng. J 2018, 332, 582–595.

Figure 6. Breakthrough curves for a binary system of Pb(II) and Ni (II) using hydrated CAM-D2EHPA as sorbent (dashed line: Thomas model; T: 20°C; internal diameter of the column, Ø: 0.5 cm; column height: 20 cm; flow rate: 0.13 mL min-1; pH: 2.2; C0: 0.24 mmol L-1).

7. The abstract was carefully reviewed, and the modifications can be found in red color in the text.

8. The following references suggested by the reviewer were added:

[21]. Ge M.; Wang, X.; Du, M.; Liang, G.; Hu, G.; S.M., Jahangir-Alam S.M.. Adsorption Analyses of Phenol from Aqueous Solutions Using Magadiite Modified with Organo-Functional Groups: Kinetic and Equilibrium Studies. Materials 2019, 12, 96.

[22]. Ge, M., Cao, L., Du, M., Hu G., Jahangir-Alam S.M., Adsorptive characterization of a pure magadiite and an organic modified magadiite on removal of methylene blue from related aqueous solution, Materials Chemistry and Physics, 2018, 217, 533-540.

The following reference was included in Table 3:

[56 ] Demey,H.; Melkior, T.; Chatroux, A.; Attar, K.; Thiery, S.; Miller, H.; Grateau, M.; Sastre, A.M.; Marchand, M. Evaluation of torrefied poplar-biomass as a low-cost sorbent for lead and terbium removal from aqueous solutions and energy co-generation. Chem. Eng. J. 2019, 361, 839–852.

Reviewer 2 Report

The paper is well presented and structured but the interaction mechanish between the sorbed heavy metals and the obtained sorben hasn't been determined.

The authors have to include this aspect in the study.

Author Response

Hary Demey

Universitat Politecnica de Catalunya

Chemical Engineering Department

Av. Diagonal, 647.

08028 Barcelona, Spain

Dr. Jun-Ichi Kadokawa,

Guest Editor of Polymers (MDPI Journal)

Dear Editor:

Please find enclosed the revised manuscript entitled Sorption and desorption studies of Pb(II) and Ni(II) from aqueous solutions by a new composite based on alginate and magadiite materials”  to Polymers (MDPI Journal). We confirm that this work has not been published, nor is it currently under consideration for publication elsewhere.

We would like to thank the referees for the careful review of the work and their useful suggestions for improving it. A thorough revision has been conducted following all their relevant considerations. This manuscript describes the manufacturing procedure of the new composite based on alginate and magadiite materials. It provides information about the immobilization of layered silicate magadiite (impregnated with D2EHPA) into calcium alginate beads. The environmentally-friendly CAM-D2EHPA material was tested for lead and nickel removal from aqueous solutions.

The CAM-D2EHPA beads were demonstrated to be promising for the selective separation of lead from synthetic effluents containing Pb(II) and Ni(II) as main pollutants (at equimolar concentration). The influences of the main parameters (i.e. pH, initial metal concentration, contact time, ionic-strength) in batch system were considered. The introduction of magadiite/D2EHPA into the alginate support, improves the metal selectivity toward Pb(II) ions.

The evaluations in the column configuration with a binary system Pb(II)-Ni(II) confirmed the reproducibility of treating effluents with a continuous process; this is particularly useful for industrial applications. The sorption-desorption cycles confirmed the easy recovery of the metals and the rapid regeneration of the sorbent with HNO3 (0.5 M) solutions was demonstrated. Also, the 10 sorption cycles allowed evaluating the mechanical stability of the CAM-D2EHPA beads; these were found to be physically stronger than the raw calcium alginate beads in presence of sodium ions. The composite CAM-D2EHPA can be used for industrial effluents, and the results obtained in this work can be extrapolated to different species of metals; this is a doorway for a new generation of green-sorbents.

As such, this paper could be of interest to a broad readership including those interested in polymer chemistry, polysaccharides applications, chemical engineering, material science and water treatment. We hope you will agree this manuscript; however, we are prepared to modify the paper according to reviewers’ comments.

Sincerely,

PhD. Hary Demey

Reviewer #2:

The paper is well presented and structured but the interaction mechanism between the sorbed heavy metals and the obtained sorbent hasn't been determined. The authors have to include this aspect in the study.

Authors:

We are very grateful to reviewer #2 for the valuable comments.

In a previous work, Attar et al [26] have shown that Na-magadiite material changes its structure after each sorption-desorption cycle in the removal of cadmium and nickel from aqueous solutions. The material becomes H-magadiite (the Na+ ions are changed by the H+ of the acid eluent).

Attar et al [26] continued their work with H-magadiite sorbent (HM) for the removal of heavy metals from aqueous solutions. In order to improve the sorption of lead and nickel ions, a dry impregnation of the silicate-based material with an acid extractant D2EHPA was carried out. This agent has been used in the solid-liquid extraction of heavy metals [23]. Figures 1.a and 1.b summarize the effect of pH on lead and nickel sorption by HM and HM-D2EHPA materials. The experiments were studied in the pH range from 1.3 to 5.7 (at diluted metal concentration: 50 mg L-1). The results obtained show an increase of the lead and nickel sorption efficiency by HM-D2EHPA reaching values of 95.5 and 27%, respectively. The decrease in equilibrium pH can be explained by the exchange between the protons of the extractant acid inserted in the powder and the metal of the solution according to the following equation:

Where M2+ is the metal cation (Pb2+ or Ni2+); HL is the acid ligand (D2EHPA) and the indices aq and org denote the aqueous and organic phases, respectively.

Further research will be developed with real effluents, which will be the scope of a future work.

Figure 1. Effect of pH on the removal of lead and nickel from aqueous solutions using: a) H-Magadiite; b) HM-D2EHPA as sorbents

The authors explained on page 9, line 326:

The lower sorption efficiency (below 50%) obtained at pH≤2.0 is partially due to the strong competition between H+ and M+2 species, which can also be related to the diffusion into actives sites of the sorbent [41]. In the same range of pH, the sorption efficiency of lead is better in comparison with nickel, this is a first indicative of the higher affinity of CAM-D2EHPA toward Pb(II) ions. The metal removal is favored at pH above 4, the sorption performance increases up to 95% for lead and 80% for nickel (Figure 3.a). This can be explained by two possible mechanisms participating: i) metal binding through carboxylic groups of the alginate, which was particularly evidenced by the competition between the metals and the protons of the solution (Figure 3.b); a buffer-effect is produced between pH 3.5-5.0. ii) Ion-exchange interactions between H-magadiite and phosphoric groups (P-OH) of D2EHPA with the metal species.

The points of zero charge (so-called pHPZC) of the sorbent and their main components were determined by the drift method [26]. Figure S2 shows that the introduction of H-magadiite impregnated with D2EHPA into the alginate matrix, contributes to reduce the pHPZC values from 6.2 to 5.9. This finding improves the metal binding of alginates and enhances the electrostatic sorbent/sorbate interactions. At pH<pHPZC, the surface of the sorbent is positively charged, this justifies the low removal of lead and nickel at acidic conditions. By increasing the pH (pH>pHPZC), the surfaces become more negative, which promotes the attraction between the sorbent and the cations metals [42].

Additional modifications:

The authors have polished the technical English, in order to provide a high quality manuscript.

The changes were highlighted (in red color) in the whole text.

Reviewer 3 Report

Comments for the manuscript entitled “Sorption and desorption studies of Pb(II) and Ni(II) from aqueous solutions by a new composite based on alginate and magadiite materials”

Dr. Keltoum Attar and coworkers report an adsorbent for removal of lead and nickel ions from aqueous solutions, which displayed either higher or comparable adsorption capacity with the other reported adsorbents in literature. The authors have given good conclusion with various experiments and they have also applied their adsorbent to dynamic system. I would like to recommend its publication in the Polymer in current form of manuscript after editing typing mistake of “HNO3” at line 135.

Author Response

Hary Demey

Universitat Politecnica de Catalunya

Chemical Engineering Department

Av. Diagonal, 647.

08028 Barcelona, Spain

Dr. Jun-Ichi Kadokawa,

Guest Editor of Polymers (MDPI Journal)

Dear Editor:

Please find enclosed the revised manuscript entitled Sorption and desorption studies of Pb(II) and Ni(II) from aqueous solutions by a new composite based on alginate and magadiite materials”  to Polymers (MDPI Journal). We confirm that this work has not been published, nor is it currently under consideration for publication elsewhere.

We would like to thank the referees for the careful review of the work and their useful suggestions for improving it. A thorough revision has been conducted following all their relevant considerations. This manuscript describes the manufacturing procedure of the new composite based on alginate and magadiite materials. It provides information about the immobilization of layered silicate magadiite (impregnated with D2EHPA) into calcium alginate beads. The environmentally-friendly CAM-D2EHPA material was tested for lead and nickel removal from aqueous solutions.

The CAM-D2EHPA beads were demonstrated to be promising for the selective separation of lead from synthetic effluents containing Pb(II) and Ni(II) as main pollutants (at equimolar concentration). The influences of the main parameters (i.e. pH, initial metal concentration, contact time, ionic-strength) in batch system were considered. The introduction of magadiite/D2EHPA into the alginate support, improves the metal selectivity toward Pb(II) ions.

The evaluations in the column configuration with a binary system Pb(II)-Ni(II) confirmed the reproducibility of treating effluents with a continuous process; this is particularly useful for industrial applications. The sorption-desorption cycles confirmed the easy recovery of the metals and the rapid regeneration of the sorbent with HNO3 (0.5 M) solutions was demonstrated. Also, the 10 sorption cycles allowed evaluating the mechanical stability of the CAM-D2EHPA beads; these were found to be physically stronger than the raw calcium alginate beads in presence of sodium ions. The composite CAM-D2EHPA can be used for industrial effluents, and the results obtained in this work can be extrapolated to different species of metals; this is a doorway for a new generation of green-sorbents.

As such, this paper could be of interest to a broad readership including those interested in polymer chemistry, polysaccharides applications, chemical engineering, material science and water treatment. We hope you will agree this manuscript; however, we are prepared to modify the paper according to reviewers’ comments.

Sincerely,

PhD. Hary Demey

Reviewer #3:

Dr. Keltoum Attar and coworkers report an adsorbent for removal of lead and nickel ions from aqueous solutions, which displayed either higher or comparable adsorption capacity with the other reported adsorbents in literature. The authors have given good conclusion with various experiments and they have also applied their adsorbent to dynamic system. I would like to recommend its publication in the Polymer in current form of manuscript after editing typing mistake of “HNO3” at line 135.

Authors:

We are very grateful to reviewer #3 for the valuable comment.

The authors corrected the typing mistake at line 135:

The preparation of H-magadiite was obtained by slow titration of Na-magadiite in suspension with HNO3 (1 M) at room temperature for 24 h under mechanical agitation (300 rpm).